# Oxygen as a Possible Technological Adjuvant during the Crushing or the Malaxation Steps, or Both, for the Modulation of the Characteristics of Extra Virgin Olive Oil

**DOI:** 10.3390/foods12112170

**Published:** 2023-05-27

**Authors:** Gabriel Beltrán Maza, Abraham M. Gila Beltrán, María Paz Aguilera Herrera, Antonio Jiménez Márquez, Araceli Sánchez-Ortiz

**Affiliations:** Andalusian Institute of Agricultural and Fisheries Research and Training (IFAPA), Agro-Industry and Food Quality Area, Center IFAPA Venta Del Llano, Mengíbar, 23620 Jaén, Spain; gabriel.beltran@juntadeandalucia.es (G.B.M.); abrahanm.gila@juntadeandalucia.es (A.M.G.B.); mariap.aguilera.herrera@juntadeandalucia.es (M.P.A.H.); antonio.jimenez.marquez@juntadeandalucia.es (A.J.M.)

**Keywords:** carotenoid pigments, chlorophyll pigments, extraction technology, olive oil quality, oxygen, phenolic compounds, tocopherols and volatile compounds

## Abstract

In commercial terms, Extra Virgin Olive Oil (EVOO) is considered an exceptional food with excellent sensory and nutritional quality due to its taste, odor, and bioactive compounds; as such, it is of great health interest. This quality can be affected by the oxidative degradation, both chemical and enzymatic (the activity of oxidative, endogenous enzymes from the polyphenol oxidase and peroxidase olive fruit type), of essential components during the extraction and conservation of EVOO. In the bibliography, oxygen reduction during the malaxation process and oil storage has been studied in different ways. However, research concerning oxygen reduction in the crushing of the olive fruit or the malaxation of the paste, or both, in the “real extraction condition” is scarce. Oxygen reduction has been compared to control conditions (the concentration of atmospheric oxygen (21%)). Batches of 200 kg of the olive fruit, ‘Picual’ cultivar, were used and the following treatments were applied: Control (21% O_2_ Mill–21% O_2_ Mixer), “IC-NM”: Inerted crushing -Normal malaxation (6.25% O_2_ Mill-21% O_2_ Mixer), “NC-IM”: Normal crushing-Inerted malaxation (21% O_2_ Mill-4.39% O_2_ Mixer) and “IC-IM”: Inerted crushing -Inerted malaxation (5.5% O_2_ Mill-10.5% O_2_ Mixer). The parameters of commercial quality covered by regulation (free acidity, peroxide value and absorbency in ultra-violet (K_232_ and K_270_)) did not suffer any change concerning the control, and so the oils belong to the commercial category of “Extra Virgin Olive Oil”. The phenolic compounds of the olives involved in the distinctive bitter and pungent taste, health properties, and oxidative stability are increased with the downsizing amounts of oxygen in the IC-NM, NC-IM, and IC-IM treatments with an average of 4, 10, and 20%, respectively. In contrast, the total amount of volatile compounds decreases by 10–20% in all oxygen reduction treatments. The volatile compounds arising from the lipoxygenase pathway, which are responsible for the green and fruity notes of EVOO, also decreased in concentration with the treatments by 15–20%. The results show how oxygen reduction in the milling and malaxation stages of olive fruit can modulate the content of phenols, volatile compounds, carotenoids, and chlorophyll pigments in the EVOO to avoid the degradation of the compound with sensorial and nutritional interest.

## 1. Introduction

In commercial terms, Extra Virgin Olive Oil (EVOO) is considered an exceptional food of excellent sensory and nutritional quality [1,2,3,4]. Its genuine spicy and bitter taste and its characteristic green-fruity odor are evaluated by trade regulations and physical-chemical parameters: acidity, peroxide value, and absorbency in ultra-violet at 232 and 270 nm. They constitute the criteria of commercial quality included in the current regulations (EEC/2568/91) [5]. Bioactive compounds present in the VOO, mainly phenolic compounds and tocopherols, have acquired great interest in health research due to their antioxidant properties included in a nutritional claim published in 2012 by the European Union: Commission Regulation (EU) n° 432/2012 [6]. Furthermore, the phenolic compounds related to bitter, spicy, and astringent attributes, typical of the VOO and its concentration in edible vegetables without refining, such as Extra Virgin Olive Oil, are considerably higher than in refined oils. These factors have contributed to the recent increase in the demand for high-quality oils [7,8].

Nevertheless, this quality may be compromised or affected by certain factors, including chemical or enzymatic, or both, oxidative degradation that can be produced during the extraction process or storage of VOO [9]. Lipid oxidation has been considered the major problem affecting edible oils, causing sensory defects, such as rancidity [10]. Simultaneously, the role of oxygen is critical in the VOO extraction process for the development of aromatic notes such as green and fruity [11,12]. These enzymatic reactions are known as the “Lipoxygenase pathway (LOX)”, which catalyze the oxidation of the 1,4-pentadiene system of polyunsaturated fatty acids, specifically linolenic and linoleic acids, to produce the corresponding hydroperoxides. These hydroperoxides are key metabolites in the pathway, as they are the initiators of both desirable or undesirable oxidation reactions [13]. The desirable evolution is produced when the lyases catalyze the hydrolysis of hydroperoxides into aldehyde compounds with six atoms (hexanal, trans-2-hexenal and cis-2 hexenal). These aldehydes are reduced by the action of alcohol dehydrogenase (ADH), forming alcohols of six atoms of carbon (hexanol, cis-2-hexenol and trans-2-hexenol) and finally, the alcohol-acetyl-transferase (AAT) catalyzes the esterification of alcohols to corresponding esters (Hexyl acetate, Z-3-hexenyl acetate and E-2- hexenyl acetate) [14]. When the substrate of the LOX enzyme is linoleic acid, saturated compounds are synthesized, however, when the substrate of the LOX enzyme is linolenic acid, unsaturated compounds are formed. An additional branch of the LOX pathway is active when the substrate is linolenic acid, through the homolytic cleavage of 13-hydroperoxides, via an alkoxy radical that gives rise to stabilized 1,3-pentene radicals. These can dimerize, leading to C10 hydrocarbons (known as pentene dimers), or couple with a hydroxyl radical in the medium to produce C5 alcohols, which can be enzymatically oxidized to corresponding C5 carbonyl compounds [15]. The importance of these volatile compounds originated by the LOX pathway for the green and fruity odor of VOO has been widely established and recently revised [1,3,4,14].

Concurrently, oxygen acts as a cofactor in the oxidation of phenolic compounds using oxidative-endogenous enzymes; in particular, for olive fruit, polyphenol oxidase (PPO) and peroxidase (POX) enzymes. Both PPO and POX activities can oxidize the main phenolic glycosides present in the fruit, especially secoiridoid compounds derived from oleuropein through the hydrolysis of oleuropein, demethyloleuropein, and ligstroside using endogenous β-glycosidases [16].

The oxidative enzymatic activities of the olives mentioned (lipoxygenase, polyphenol oxidase, and peroxidase) are activated during the VOO extraction process. In particular, during the crushing process, they are activated after their release, owing to the cellular disruption of fruits and during the malaxation phase-partition phenomena between oil and water, and vice versa, and are responsible for the change in the composition of VOO (1).

The role of oxygen during the olive oil extraction process has been analyzed from various perspectives in the last decade. In particular, most of this research has focused on the malaxation phase at a laboratory and industrial scale and, in Italian cultivar, mainly ‘Moraiolo’ and ‘Frantoio’. The modification of the oxygen concentration in the headspace of the malaxer to improve the quality of VOO has been tested with different devices and approaches [12,17,18,19,20,21,22,23,24,25,26,27]. In general, the decrease in oxygen during malaxation reduces the oxidation of phenolic compounds [20,21,22,26] and chlorophylls [25]. However, the decrease in oxygen during malaxation is not significant for volatile compounds [12,20,22]. Studies that increase the concentration of oxygen during the malaxation step have enhanced the concentration of volatile compounds [11,18,27,28]. The combining of high-power ultrasound pre-treatment with malaxation oxygen control to improve the quantity and quality of Extra Virgin Olive Oil has been published by Iqdiam et al. [29]. Most recently, the development of a malaxer with a supervisory control and data acquisition system (SCADA) for oxygen and process duration monitoring made it possible to increase the values of the tocopherols and total phenol content [30]. In addition, inert gases such as argon, nitrogen, and carbon dioxide have been applied in the vertical centrifugation and storage process to ensure optimal preservation during the oil extraction and its storage over time [31,32].

Nevertheless, the crushing of the fruit has been poorly studied compared to the malaxation of the olive paste. Once the fruit is crushed, metabolic pathways related to the volatile and phenolic compounds are activated; their control during both steps, crushing and malaxation, is crucial for the final quality of the EVOO. The role of oxygen during crushing on volatile, phenol, and sensory properties has been stated. In particular, the increase in oxygen during the crushing has been described by research [11,28,29,30,31,32,33] and reduced by Vezzaro et al. [19] and Sánchez-Ortiz et al. [12]. The present work aims to investigate the role of oxygen, not only on volatile compounds and phenols, but also on quality parameters and other relevant minority compounds such as chlorophylls, carotenoid pigments, and tocopherol during crushing and malaxation at an industrial scale in ‘Picual´ cultivar, one of the most widely distributed strains in the world. The novelty of this work lies in the study at an industrial level of the crushing and malaxation steps. For this purpose, the atmospheric oxygen (21% O_2_) during the crushing of fruit and the malaxation of the paste has been reduced by the application of nitrogen and the sealing of the mill and mixer. The results obtained are key for the development of new technological strategies for the modulation of the EVOO characteristics depending on the target markets. The development of this type of strategy is fundamental because the olive oil extraction process is subject to precise regulations that only allow the use of oxygen, water, and talc as technological coadjuvants.

## 2. Materials and Methods

### 2.1. Olive Fruits

Olive fruits of the cultivar ‘Picual’ from a traditional olive grove were harvested in the experimental farm of the IFAPA Center “Venta Del Llano” in Mengíbar (Jaén), Spain. Olive fruits (1000 kg) were collected with trunk shakers in early and mid-December 2021. They were named trial 1 and trial 2. The olive fruits were separated into boxes of 25 kg and were processed within the next few hours (between 4–5 h). The characteristics of the olive fruit have been analyzed following “The guide for the determination of the characteristics of oil-olives”, published by the International Olive Council in COI/OH/Doc. No 1/2011 November 2011 [34].

### 2.2. Olive Oil Extraction and Treatments

For the olive oil extraction in control conditions (21% O_2_) with oxygen reduction (inertization) in crushing or malaxation, or both, a system of extraction of the two phases with a hammer mill, horizontal mixer (350 l), and centrifugal extractor was used (Model Il Molinetto, Pieralisi, Italy). The modification of the atmospheric oxygen concentration in the milling and malaxation of the olive paste was performed by introducing molecular nitrogen into the mill and mixer, previously sealed with polypropylene. The percentage of oxygen was monitored with a device called Oxybaby every ten minutes (WITT Gas, Witten, Germany). For each extraction trial, batches of 200 kg of olive fruit were processed. Figure 1 shows in parentheses the average level of oxygen in atmospheric conditions and the measures for each treatment and repetition (A and B) in milling and malaxation. The open-air step is considered “normal or control”. For each extraction trial, the olive paste was mixed for 30 min at 25 °C and the oils obtained after centrifugation were decanted, filtered, and stored under frozen conditions (−20 °C) until analysis.

### 2.3. Chemicals

All solvents and reagents used were of HPLC grade or equivalent. α-, β-, and γ-Tocopherol, caffeic acid and standards for volatile compounds (Appendix A) were purchased from Merck (Germany).

### 2.4. Quality Parameters Analysis

Free acidity, peroxide value, and absorbency in ultra-violet at 232 and 270 nm were determined according to the European Union Commission Regulations EEC/2568/91 and amendments with the international referee methods described by the International Olive Council in COI/T.15/NC No 3/Rev. 16 June 2021 [35].

The content of free fatty acids was expressed as free acidity, calculated as the percentage of oleic acid. The samples were dissolved in a mixture of diethyl ether and ethanol, and the free fatty acids were titrated using a potassium hydroxide solution. The peroxide value was evaluated by titration and absorbency in ultra-violet light at 232 and 270 nm, which measures the presence of the conjugated diene and triene systems resulting from oxidation or refining, respectively.

### 2.5. Minor Compounds

Tocopherols were determined according to the IUPAC method N° 2.436 (1992) [36] with slight modification. Sample tests were dissolved with propan-2-ol in hexane (0.5/99.5% *v*/*v*) and tocopherols were determined directly by HPLC analysis using an Agilent 1200 series equipped with an analytical column Lichrosphere Si 60 (Merck) 250 mm × 4.6 mm, with 5 µm of mean particle size and a UV detector. The wavelength of the UV detector set was 292 nm. The quantification was carried out by calibration factors determined for each tocopherol from the chromatography of the solutions of the standard tocopherols described in Section 2.3.

Chlorophylls and carotenoid pigments analyses were determined according to the protocol established by Mínguez-Mosquera et al. (1991) [37] using a UV-Visible spectrophotometer (Varian Cary 50 Bio) at 472 nm for the carotenoids and 670 nm for the chlorophylls with oil samples dissolved in cyclohexane (0.15 g × mL^−1^).

A total phenols content analysis was based on the Folin-Ciocalteu reagent per Vázquez-Roncero et al. (1973) [38]. This colorimetric method is broadly applied for the determination of phenols using an aqueous methanol extract at 60% (*v/v*). The content was expressed in mg × kg^−1^ of caffeic acid.

The analysis of the volatile compounds was based on the protocol published by Sánchez-Ortiz et al. 2018 [39]. The extraction of the volatiles was performed by a Solid Phase Micro Extraction with a fiber of divinylbenzene/carboxen/polydimethylsiloxane (DVB/CAR/PDMS) 50/30 µm (Supelco Co., Bellefonte, PA, USA). The volatiles absorbed were separated and identified on a Bruker model Scion 456-GC-TQMS system (Bruker, MA) equipped with a Supelcowax 10 capillary column (60 m × 0.25 mm i.d.; thickness, 0.25 µm; Sigma-Aldrich Co. LLC). The volatile compounds were identified at different levels following the criteria defined by the metabolomics-standard initiative [40]. In Appendix A, the metabolites were definitively annotated (level 1) by comparing the MS spectra and linear retention index (LRI) against the available standards purchased from Sigma-Aldrich-Merck. Putative (or tentative) identifications (levels 2 and 3) were considered by comparing the MS spectra and LRI against existing databases (NIST 17 v2.3). Metabolites for the MS spectra and LRI that were not available in the bibliography were labelled as “unknown” (level 4). Volatile compounds were quantified by the calibration lines with standards (level 1), and other volatile compounds were normalized by internal standards (levels 2, 3 and 4).

### 2.6. Statistical Analysis

The data obtained in triplicate were expressed as mean ± standard deviation. An analysis of variance was carried out to determine whether the reductions in oxygen affect the results obtained in the control and treatments: T1, T2, and T3 for each trial A and B. For this purpose, Tukey’s method was used with Statistix v.9. Moreover, to extract meaningful information from volatile compounds the Principal Compound Analysis (PCA) was applied to the data matrix using MetaboAnalyst 5.0.

## 3. Results and Discussion

### 3.1. Quality Parameters

The data obtained for the characterization of the olive fruit used for the extraction of the EVOO in this study are shown in Table 1. Maturity index, weight medium, FMO (total oil content on a fresh matter basis), DMO (total oil content on a dry matter basis), and moisture of the cultivar ‘Picual’ in two different harvesting dates showed similar data in trial A and B, with an average for both test of 3.56, 2.74, 54.51, 19.01, and 54.49, respectively. The results are consistent with the variety and the time of harvesting of the fruit [41]. 

Concerning the commercial quality included in the regulation, all data are displayed in Table 2: free acidity, peroxide value, and absorbency in ultra-violet at 232 and 270 nm. The free acidity was between 0.18 and 0.24; these values match with the “Extra Virgin Olive Oil (EVOO)” commercial grade (≤0.80% in oleic acid). Peroxide value, expressed as milliequivalents of active oxygen per kilogram (mEq O_2_ per kg) of oil, did not exceed 5.56 in any sample, corresponding with the high commercial quality “EVOO” (≤20.0 mEq O_2_/kg). The limits for absorbency in ultra-violet at 232 and 270 nm for the EVOO are ≤2.50 and ≤0.25, respectively. The maximum experimental values for these parameters were 1.79 in T1 trial B for 232 nm and 0.18 in T2 and T3 in trial B for 270 nm. Both levels correspond with EVOO grades. Peroxide value and absorbency in ultra-violet at 232 and 270 nm evaluated the level of oxidation of the oil. The decrease in oxygen in the different treatments applied in the oil extraction produced no effect on the parameters of oxidation. Parenti et al. [24], in cultivar Frantoio on a laboratory scale, reported how the effect of blanketing with CO_2_ during malaxation did not produce significant differences in acidity and K_270_; however, oil produced in SC (Sealed conditions) showed a lower PV (Peroxide Value) and K_232_.

### 3.2. The Effect of the Treatments of Reduction of Oxygen on the Minor Fraction of Extra Virgin Olive Oil

As mentioned earlier, tocopherols, phenols, volatile compounds, chlorophylls, and carotenoid pigments are closely linked to the sensory, biological, and technological Extra Virgin Olive Oil properties. Tocopherols constitute a key group of antioxidant compounds; their levels may vary between 70 and 600 mg × kg^−1^ in VOO [42]. Table 3 shows an average total content in the four treatments of 285.95 and 373.93 mg × kg^−1^ for trials A and B, respectively. Three isoforms of tocopherols were identified in the VOOs: α-, β-, and ɣ-tocopherol. α-Tocopherol represents more than 90% of the total and β- and ɣ-tocopherol, 2% and 5%, respectively. The analysis of tocopherols revealed that oxygen reduction does not significantly affect the content of these compounds in trial B. In trial A, a very slight decrease (minus 4%) for T1 compared to the TC and a very slight increase (1%) for T2 compared to the TC, were shown. It has been described as the extraction technology marginally affecting the tocopherol concentration [43,44].

Concerning the pigments occurring in the VOO responsible for its characteristic colour, carotenoid and chlorophyll pigments have been analyzed in the present study (Table 4). A slight rise can be observed in the content of the pigments, from inertization VOOs (T1, T2, and T3) with respect to the control treatment (TC). The level of pigments in VOOs depends mainly on the cultivar, ripeness, irrigation condition of the olive fruit, and the conditions of the extraction process [45]. When the malaxation phase was carried out through blanketing with CO_2_, Parenti et al. [24,25] reported a higher chlorophyll and phenols content than in the oil produced in sealed conditions with respect to the control (open-air malaxation), confirming a similar effect.

During the VOO extraction process, compounds, such as phenolic and volatile compounds, are transformed due to biochemical reactions triggered by the crushing of the olive fruit. In the case of phenolic compounds, oleuropein is mainly hydrolyzed by β-glucosidase, giving rise to more lipophilic secoiridoid derivatives [16]. These compounds, along with the tocopherols, are key antioxidant compounds of the oxidative stability of the EVOO. Moreover, together with volatile compounds, they are responsible for the positive organoleptic characteristics, such as pungent, bitter, and fruity. Specifically, the average total phenol content in the treatments (TC, T1, T2, and T3) in Figure 2 was between 311 and 613 mg × kg^−1^ of oil for trials A and B, respectively. As the data show, the effect of oxygen reduction during the global extraction process (T3) with respect to the control (TC) is statistically significant, with an increase of 23% and 13% for trials A and B, respectively. As described in the introduction, the role of oxygen during the malaxation step on phenolic fraction has been established in previous studies to reduce the oxidation and damage of these compounds by endogenous oxidoreductases, such as polyphenol oxidase and peroxidase. When the oxygen is limited in the malaxation step, an increase in phenol content has been reported in all the literature revised at laboratory, pilot, and industrial scale [20,21,22,23,24,26,27], increasing this parameter by 50 percent in the work published by Parenti et al. [25]. Unlike these studies, in the present study, the effect of the blanketing with nitrogen has been analyzed either on the whole process or within the milling or malaxation steps. The highest effect (23%) on total phenols was observed when the conditions of oxygen reduction were performed over the whole process (crushing and malaxation: T3) with respect to the control (TC). These results suggest that the treatment could increase oxidative stability and sensory properties, such as bitter and pungent.

A contrary effect was found for the volatile compounds. The final group of compounds analyzed during this study is related to the genuine and characteristic aroma of the EVOO. Their biosynthesis through the lipoxygenase (LOX) pathway during the extraction process requires oxygen to catalyze the oxidation of free polyunsaturated fatty acid bearing the 1-cis, 4-cis-pentadiene system and, consequently, the restriction of oxygen could compromise the biosynthesis of C6 and C5 compounds derivatives of the LOX pathway, which relates to the green and fruity notes of EVOO. In fact, as shown in Figure 3, the LOX derivative volatile compounds from the inerting treatments (T1, T2, and T3) are reduced by 15–20% regarding the control in the A and B trials, respectively. Similar reductions have been described in treatments with a decrease in oxygen in the malaxation phase [26]. In contrast, aroma biosynthesis during malaxation was minimally affected by oxygen reduction in Servili et al. [22] and Masella et al. [20]. These different results could be linked to the varietal difference and the ripening stage used in these studies, as reported by Sánchez-Ortiz et al. [11,12]. In addition, regarding LOX-derivative volatile compounds, it should be noted that there are no significant differences between the inertization treatments: T1, T2, and T3 (Figure 3).

The effect of oxygen on all volatile compounds identified in this study (LOX derivatives and no LOX derivatives compounds) was analyzed by multivariate analysis: Principal Compound Analysis (PCA) model. Figure 4 shows a bi-plot of the first two principal components (PC1 and PC2) with the scores (on the top) and loadings (on the bottom) for the A and B trials. In particular, 27 compounds were identified (determined) and grouped by their chemical nature in sum: furans, non-LOX aldehydes, non-LOX alcohols, non-LOX ketones, non-LOX esters, aromatic hydrocarbons, terpenes, un-identified, LOX-aldehydes, LOX-alcohols, LOX-esters, LOX-hydrocarbons, and LOX esters. The level of identification is shown in Appendix A. The first principal component retains 50.6% and 28.9% of the original variance, and the second principal component retained 28.9% and 15.2% of the original variance for the A and B trials, respectively. The different treatments of inertization (TC, T1, T2, and T3) were separated by a second component in score plots “a” and “b” from Figure 4 for both tests. Therefore, the control treatment was related to LOX aldehydes in loading and score plots. Likewise, treatment 2 was related to Aromatic hydrocarbons and LOX-esters. Finally, treatments 1 and 3 were related to other groups of volatile compounds, such as No LOX esters and Aromatic hydrocarbons. The data show LOX aldehydes as a possible marker of oxygen reduction. The LOX aldehydes are related to the green odor of high-quality EVOO [1,3].

In conclusion, the decrease in oxygen in the different treatments applied in the crushing and the malaxation during the EVOO extraction did not affect the parameters of quality or tocopherols. Therefore, the decrease in oxygen in the overall process, or within each stage of the extraction process, could be used to modulate the content of phenol, pigments and volatile compounds. All of them are responsible for the sensorial properties (colour, taste and aroma) of high-quality EVOO depending on our target market. The results obtained are key for developing new technological strategies for the modulation of the EVOO characteristics.

## Figures and Tables

**Figure 1 foods-12-02170-f001:**
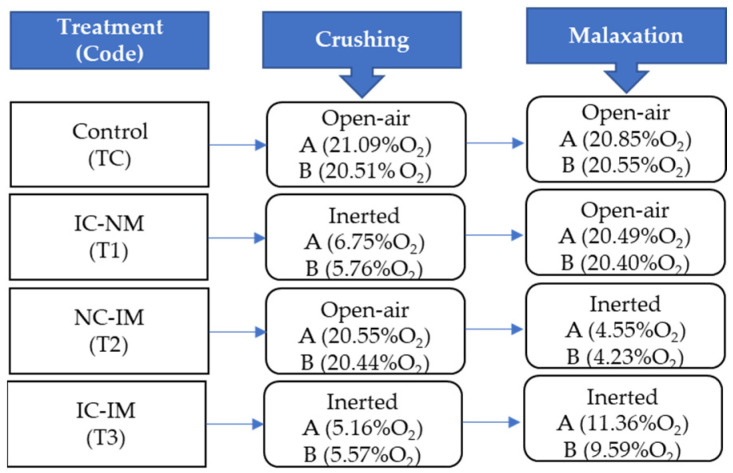
The average percentage of oxygen in the experimental design for each treatment (TC, T1, T2, T3) used in the study during the crushing of the olive fruit and the malaxation of the olive paste in the A and B trials.

**Figure 2 foods-12-02170-f002:**
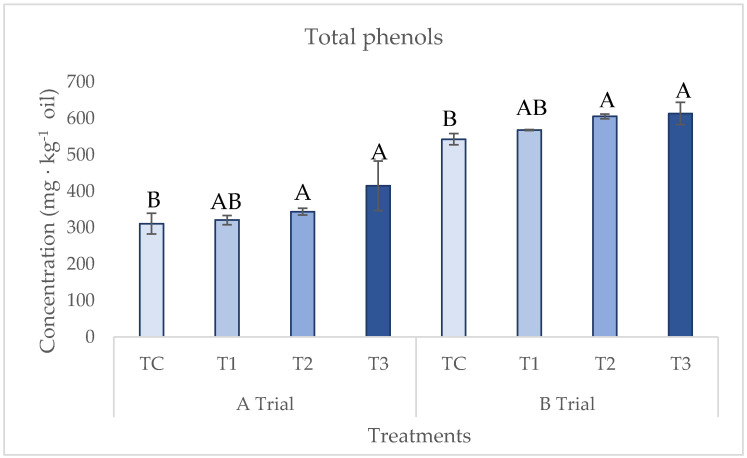
The content of total phenols (mg × kg^−1^ oil) for the inertization treatments, as described in Figure 1. Significant differences between the treatments (TC, T1, T2, and T3) are shown with different letters for *p* < 0.05 within each A and B trial (ANOVA Test).

**Figure 3 foods-12-02170-f003:**
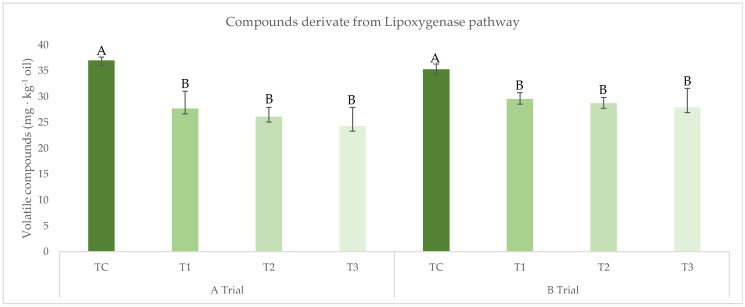
The content of volatile compounds (LOX derivative) (mg × kg^−1^) for the inertization treatments, as described in Figure 1. Significant differences between treatments (TC, T1, T2, and T3) are shown with different letters for *p* < 0.05 within each A and B trial (ANOVA Test).

**Figure 4 foods-12-02170-f004:**
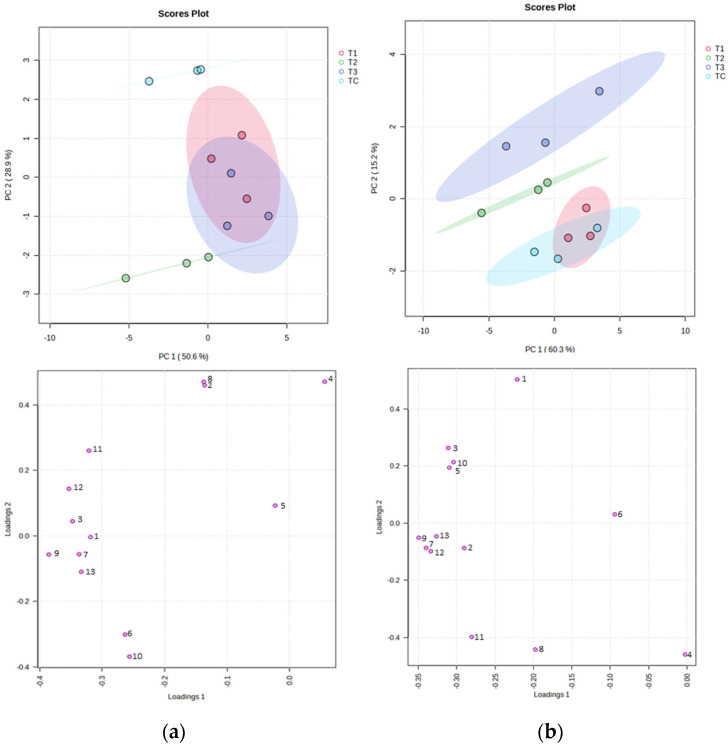
The Principal Compound Analysis (PCA) of the first two principal components (PC1 and PC2) in the treatments applied (TC, T1, T2, and T3) for trial A (**a**) and trial B (**b**). A group label corresponding to the volatile compounds classified and identified in Appendix A in 13 groups of compounds: 1. Furans, 2. No LOX aldehydes, 3. No LOX alcohols, 4. No LOX ketones, 5. No LOX esters, 6. Aromatic hydrocarbons, 7. Terpenes, 8. LOX aldehydes, 9. LOX alcohols, 10. LOX esters, 11. LOX hydrocarbons, 12. LOX Ketones, and 13. Unidentified.

**Table 1 foods-12-02170-t001:** The characterization of the fruit ‘Picual’ cultivar used in the tests A and B. HD (Harvest date), MI (maturity index), WM (Weight medium), FMO (total oil content on a fresh matter basis), DMO (total oil content on a dry matter basis), and Moisture.

Trial	HD	MI	WM (g)	FMO (%)	DMO (%)	Moisture (%)
A	03/12/2021	3.76	3.09	19.45	56.82	43.18
B	15/12/2021	3.36	2.40	18.58	52.20	47.80

**Table 2 foods-12-02170-t002:** The quality parameters determined in A and B trials, as described in Figure 1, for each treatment of oxygen reduction (TC, T1, T2 and T3). (*) Free acidity in % m/m expressed in oleic acid. Peroxide value expressed in milleq. peroxide oxygen per kg/oil. Absorbency in ultra-violet at 232 and 270 nm calculated for (K^1%^)_1cm_.

Parameter *	A Trial			
	TC	T1	T2	T3
Free acidity	0.22 ± 0.03	0.20 ± 0.03	0.18 ± 0.00	0.20 ± 0.03
Peroxide value	4.56 ± 0.52	4.73 ± 0.25	4.25 ± 0.29	4.24 ± 0.29
K_232nm_	1.70 ± 0.11	1.64 ± 0.04	1.58 ± 0.01	1.65 ± 0.03
K_270nm_	0.15 ± 0.04	0.11 ± 0.03	0.10 ± 0.01	0.13 ± 0.01
	**B Trial**			
	TC	T1	T2	T3
Free acidity	0.24 ± 0.00	0.23 ± 0.00	0.22 ± 0.00	0.21 ± 0.00
Peroxide value	5.56 ± 0.00	4.99 ± 0.02	5.17 ± 0.63	4.98 ± 0.02
K_232nm_	1.75 ± 0.02	1.79 ± 0.06	1.76 ± 0.03	1.74 ± 0.02
K_270nm_	0.15 ± 0.00 *	0.16 ± 0.01	0.18 ± 0.01	0.18 ± 0.01

ANOVA Test. Significant differences between treatments (TC, T1, T2 and T3) are shown with an asterisk (*) for *p* < 0.05 within each A and B trial.

**Table 3 foods-12-02170-t003:** The content of α-tocopherol, β-tocopherol, and ɣ-tocopherol in mg × kg^−1^ of oils in different treatments of inertization (TC, T1, T2 and T3) for A and B trials according to Figure 1.

	A Trial			
	TC	T1	T2	T3
α-tocopherol	257.24 ± 8.11 bc	254.93 ± 1.62 c	268.74 ± 4.63 ab	271.57 ± 4.33 a
β-tocopherol	6.34 ± 0.33 a	6.05 ± 0.04 a	6.19 ± 0.45 ab	6.23 ± 017 a
ɣ-tocopherol	16.51 ± 0.68 a	16.07 ± 0.07 a	17.00 ± 0.44 ab	16.95 ± 0.31 a
Total	280.10 ± 9.09 ab	277.05 ± 1.65 b	291.92 ± 4.72 ab	294.75 ± 4.61 a
	**B Trial**			
	TC	T1	T2	T3
α-tocopherol	352.46 ± 2.62 a	356.94 ± 1.84 a	357.46 ± 2.41 a	355.55 ± 2.54 a
β-tocopherol	6.81 ± 0.17 a	6.84 ± 0.20 a	6.77 ± 0.04 a	6.96 ± 0.04 a
ɣ-tocopherol	14.39 ± 0.12 a	14.80 ± 0.52 a	14.58 ± 0.12 a	14.14 ± 0.05 a
Total	373.66 ± 2.83 a	378.59 ± 2.26 a	378.82 ± 2.36 a	376.66 ± 2.56 a

ANOVA Test. Significant differences between treatments (TC, T1, T2 and T3) are shown with different letters for *p* < 0.05 within each A and B trials.

**Table 4 foods-12-02170-t004:** The content of carotenoid and chlorophyll (mg × kg^−1^ of oil) in different treatments of inertization (TC, T1, T2 and T3) for A and B trials, as described in Figure 1.

	A Trial			
	TC	T1	T2	T3
Carotenes	2.16 ± 0.05 c	2.98 ± 0.17 a	2.93 ± 0.08 a	2.52 ± 0.11 b
Chlorophylls	1.21 ± 0.10 a	1.73 ± 0.20 a	1.61 ± 0.23 a	1.33 ± 0.26 a
	**B Trial**			
	TC	T1	T2	T3
Carotenes	7.94 ± 0.11 b	8.36 ± 0.16 ab	8.72 ± 0.14 a	8.77 ± 0.36 a
Chlorophylls	6.39 ± 0.18 c	7.32 ± 0.32 b	8.15 ± 0.11 ab	8.81 ± 0.54 a

ANOVA Test. Significant differences between treatments (TC, T1, T2 and T3) are shown with different letters for *p* < 0.05 within each A and B trials.

## Data Availability

The data generated in this study can be provided by the corresponding author upon request.

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
