# Peer review of "Oxygen as a Possible Technological Adjuvant during the Crushing or the Malaxation Steps, or Both, for the Modulation of the Characteristics of Extra Virgin Olive Oil"

_foods, 2023, doi:10.3390/foods12112170_

Round 1

Reviewer 1 Report

Dear Authors, as correctly pointed out, the use of nitrogen to reduce the amount of oxygen during the malaxation phase has long been studied. As a reviewer, however, I appreciated that this particular work also focuses on the crushing phase. The study is supported with evidence, however I have some questions and observations:

1. Does the research concern VOO or EVOO? This aspect is crucial but also not very clear because, in my opinion, the use of nitrogen would make sense for the extraction of EVOO. I, therefore, invite the authors to make this aspect clear (eventually also in the title).

2. Are there any downsides to using nitrogen? (e.g. increased cost, safety and quality concerns, increased system complexity, etc.).

3. Has a drop in temperature been observed after the introduction of nitrogen?

4. Please, insert names in italics or in quotation marks.

5. Please, enter keywords in alphabetical order.

6. Please check punctuation and put the bibliography in alphabetical order.

7. I recommend a final paragraph for conclusions only.

I suggest the authors a thorough double check of the work in order to improve English and overcome several errors related to spelling, punctuation, upper and lower case, bibliography references in the text, unnecessary repetitions etc. 

Reviewer 2 Report

the authors discuss the effects, on the quality and composition of the extra virgin olive oil, of the control of the oxygen content during the milling and malaxation phases. In the literature, as the authors also point out, there are case studies. The added value, according to the authors, of this work is due to the verification of the effects of a simultaneous control, in the two phases, of oxygen. The work flows smoothly, its experimental setup is satisfactory but needs some minor adjustments.

 I suggest some improvements listed below:

 1)     I suggest comments in the text to explain the comparison performed and indicated in tables and figures as “Significant differences between treatments are shown with asterisk (*) for 239 p<0.05” or “Significant differences between treatments are shown with different letters for p<0.05”. The sentences are ambiguous, it would be advisable to indicate which groups of data are compared and better explain which comparisons the letters or symbols are associated with.

2)     in line 20-23: indicate the coding of treatments as “Control (21% O2 Mill - 21% O2 Mixer), “IM-NM”: Inerted 20 milling -Normal malaxation (6.5% O2 Mill-21% O2Mixer), “NM-IM”: Normal milling-Inerted malax- 21 ation (21% O2 Mill-4 % O2 Mixer) and “IM-IM”: Inerted milling-Inerted malaxation (5.5% O2 Mill- 22 10% O2 Mixer)” but in the following text different codes are used. It needs homogenization of codes all over the text, pictures and tables included. I suggest to add a coherent code also for the control.

3)     Line 25: “comercial” I suppose is “commercial”

4)     Lines 117, 225, 227, 238, 239, 240, 427:  O2” in place of “O2

5)     Lines 425:  “CO2” in place of “CO2

6)     Line 179: “g * mL-1” would be “g * mL-1”. The measurements units are written, all over the text, in different ways. I suggest a unique symbolism for each unit of measurement.

7)     Line 183: “mg per kg-1 “ I think it more consistent with the other units if written “mg * Kg-1”. Use similar cure for other units.

8)     Pag. 4 figure 1: numbers must be written in Anglo-Saxon notation

9)     Pag. 6 table 2 and 3, pag. 7 table 4, pag. 8 Figure 2 pag.9 figures 3 and 4: There are codes for treatments never explained in the text. Make them consistent.

10)  Line 247: the values indicated are not consistent with those in table 3.

11)  Pag. 9 figure 4: the figure must be commented in the text describing the data analyzed and the value of the result. I think even its caption must be simplified.
